# Characterization of Key Factors Associated with Flavor Characteristics in Lager Beer Based on Flavor Matrix

**DOI:** 10.3390/foods14101702

**Published:** 2025-05-11

**Authors:** Jiaxin Hong, Huayang Wei, Ruiyang Yin, Jiang Xie, He Huang, Liyun Guo, Dongrui Zhao, Yumei Song, Jinyuan Sun, Mingquan Huang, Baoguo Sun

**Affiliations:** 1China Food Flavor and Nutrition Health Innovation Center, Beijing Technology and Business University, Beijing 100048, China; m15205199608@163.com (J.H.); 13563938331@163.com (H.W.); 18810093270@163.com (J.X.); huanghe3938@163.com (H.H.); sunjinyuan@btbu.edu.cn (J.S.); huangmq@th.btbu.edu.cn (M.H.); sunbg@btbu.edu.cn (B.S.); 2Key Laboratory of Brewing Molecular Engineering of China Light Industry, Beijing Technology and Business University, Beijing 100048, China; 3Beijing Laboratory of Food Quality and Safety, Beijing Technology and Business University, Beijing 100048, China; 4Department of Nutrition and Health, China Agriculture University, Beijing 100048, China; 5Beijing Key Laboratory of Beer Brewing Technology, Technology Center of Beijing Yanjing Beer Co., Ltd., Beijing 101300, China; ruiyang_yin@163.com (R.Y.); yanjing6089@163.com (L.G.); songym@163.com (Y.S.); 6School of Food and Health, Beijing Technology and Business University, Beijing 100048, China

**Keywords:** lager beer, flavoromics, flavor compounds, sensory evaluation, validation experiments

## Abstract

Lager beer has the characteristics of a refreshing aroma, clean and less intense taste, as well as a low alcohol degree, which is suitable for daily drinking. This study aimed to clarify the relationship between important flavor compounds and flavor profiles for lager beer. Headspace solid-phase microextraction, solvent-assisted flavor evaporation combined with comprehensive two-dimensional gas chromatography–mass spectrometry, gas chromatography–mass spectrometry, and gas chromatography-olfactometry–mass spectrometry were applied for the qualitative and quantitative analysis of the trace components in lager beer. Furthermore, the recombination experiment was successfully applied to simulate the flavor profile, and the omission experiment was conducted to study the effects of flavor compounds on the flavor profile. A total of nine compounds were identified as the key flavor compounds, and their contribution to the flavor characteristics of lager beer was verified according to validation experiments. It was found that the influence of the key flavor compounds on the sensory attributes such as malty aroma, fruity aroma, sweetness, and bitterness varied with their concentration. These findings might provide ideas for the research regarding the flavor compounds and flavor profile of lager beer, and contribute to the development of different types of beer in the future.

## 1. Introduction

As one of the important alcoholic beverages in the daily lives of people, the history of beer can be traced back to 13,000 years ago [1]. The early development of the Chinese beer industry was relatively slow. However, since the 21st century, the Chinese beer industry has experienced rapid development due to the improvement of technology, funding, and other factors. Nowadays, China has become one of the major producers and consumers of beer in the world. According to the data of National Bureau of Statistics of China (https://www.stats.gov.cn/), in 2023, the annual yield of the beer industry in China reached 37.89 million kiloliters, and the total sales revenue was 186.3 billion yuan (Figure 1). Moreover, the total annual profit reached 26 billion yuan. Of note, several brands of beer are well known in China including Snow, Tsingtao, and Yanjing, with a total market share of over 50%.

As a kind of alcoholic beverage, the flavor of beer is considered the primary factor influencing consumer trends. Meanwhile, the flavor profiles of beer are highly discernable by consumers based on their flavor (taste and aroma) and mouthfeel. Most beers are divided into Ale beer and lager beer. Broadly speaking, Ale beer is mainly fermented at room temperature, and the fermentation time is relatively short, while the yeast used in lager beer needs a cooler temperature, and lager beer needs a period of maturity after the main fermentation [2]. Among them, lager has a refreshing aroma, a clean and less intense taste, as well as a low alcohol degree, which is suitable for consumers’ daily drinking. The flavor characteristics and quality of beer mainly depend on the trace components in it [3,4,5]. Several trace components are found in beer, including amines, sulfur-containing compounds, alcohols, phenols, aldehydes, acids, ketones, esters, etc. [6,7,8,9]. At present, lots of high-precision detection instruments have been applied to the study of the trace components in beer, providing more efficient technical means for the research on beer flavor [10]. These detection instruments mainly include gas chromatography–mass spectrometry (GC-MS) [11,12], gas chromatography-olfactometry–mass spectrometry (GC–O–MS ) [13], comprehensive two-dimensional gas chromatography–mass spectrometry (GC×GC-MS) [14], flavor analysis systems, and so on. Meanwhile, headspace solid-phase microextraction (HS-SPME), liquid–liquid extraction (LLE), solvent-assisted flavor evaporation (SAFE), which are regarded as pre-treatment techniques, are increasingly being applied in the detection of the flavor compounds in beer [15]. Nowadays, the research on the flavor of beer is mainly focused on analyzing the relationship between the trace components and formation of the flavor characteristics associated with brewing raw materials, the fermentation conditions, and yeast strains [16,17,18,19,20,21,22]. In 2022, aroma extraction dilution analysis (AEDA) and the calculation of the odor activity value (OAV) were applied to identify the impact of caramel and roasted wheat malts on the aroma compounds in top-fermented wheat beer [23]. In 2023, the starter *Streptococcus thermophilus* TH-4 (TH-4) and the probiotics *Lacticaseibacillus paracasei* F19 and 431, associated with *Saccharomyces cerevisiae* US-05 were found to contribute to the beer flavor and were recommended for applications in sour beers [24]. Recent research has also highlighted the changes in volatile compounds and the formation of staling compounds during beer storage [25]. Meanwhile, many researchers have also focused on the influence of flavor compounds on the flavor profiles of beer. In 2019, SPME and the gas chromatography-mass-selective detector (GC-MSD) were applied to analyze the trace components in beer. Meanwhile, multiple regression analysis, and an artificial neutral network model (ANN) were developed with the peak areas of 10 flavor compounds to evaluate the flavor profile and overall liking of beer [3]. Compared with other types of beer, less research has been conducted on flavor compounds that affect the sensory attributes of commercial lager.

This study aimed (1) to unravel the flavor compounds in lager beer via modern flavoromics, (2) to confirm and compare the expression of flavor compounds, (3) to clarify the relationship between the flavor compounds and sensory attributes of lager beer by using a flavor matrix, and (4) to further verify the key chemical factors with different concentrations that affect the sensory attributes and flavor profile via validation experiments.

## 2. Materials and Methods

### 2.1. Sample Preparation

Three lager beer samples including A, B, and C were used in this study. These samples were purchased from Yanjing (Beijing, China), Huarun (Shanghai, China), and Tsingtao Industry Co., Ltd. (Qingdao, China), and were stored at 4 °C before analysis. Their raw materials were water, malt, rice, hops, and yeasts. Their alcohol contents were 4%, 3%, and 4%, respectively.

### 2.2. Chemicals

Ethanol and dichloromethane were of HPLC grade with a purity of 99.9% and purchased from Aladdin Bio-Chem Technology Company (Shanghai, China).

The internal standards (IS) including pentyl acetate (IS1), 4-octanol (IS2), and 2-ethylbutyric acid (IS3) with at least 97% purity were used for the quantification and validation experiments. They were purchased from J&K Scientific Company (Beijing, China).

### 2.3. Pretreatment with LLE, SAFE, and HS-SPME

LLE: Salt was added to the 150 mL sample, and extracted three times using dichloromethane (50 mL each time).

SAFE: A 500 mL round-bottom flask was placed as the receiving bottle, a cold trap was filled with liquid nitrogen, and the other side was put in a 40 °C water bath. The composite turbo molecular pump was turned on. Then, the LLE extracts of beer samples were slowly poured into the drip funnel when the absolute pressure of the system reached 1 × 10^−4^ MPa, starting the extraction. Particularly, the flow rate of the sample was controlled at a constant speed. The receiving flask was then removed, and the sample was allowed to melt naturally at room temperature. The sample was collected and concentrated to 1 mL.

HS-SPME: A total of 5 mL of beer sample was added to a 20 mL headspace vial, along with 1.25 g of NaCl and 25 μL of internal standard. The mixture was equilibrated at 45 °C for 10 min, followed by adsorption of the 50/30 µm DVB/CAR/PDMS fiber at the same temperature for 40 min. After extraction, the fiber was inserted into the injection port for desorption for 5 min at a temperature of 250 °C, and then analyzed via GC-MS.

### 2.4. Identification and Analysis of the Flavor Compounds

GC × GC–MS was used to qualitatively identify the trace components in lager beer. Two DB-17MSs (1.0 m × 0.15 mm × 0.30 µm) connected in series were used as the columns. The oven temperature was held at 50 °C for 3 min at first, then raised to 240 °C at a rate of 6 °C/min and held for 10 min. The inlet hot zone temperature of the modulator was set to an initial temperature of 110 °C, maintained for 3 min, and then raised to 300 °C at a rate of 6 °C/min, maintained for 10 min. The output hot zone of the modulator was 170 °C, maintained for 3 min, then raised to 320 °C at 6 °C/min and maintained for 15 min.

The MS was detected in an electron ionization (EI) mode at 70 eV. The temperature of the ion source was set at 230 °C. The mass range was set from 45 to 350 amu.

GC–O–MS was utilized to identify the flavor compounds. A polar chromatographic column DB-WAX (60 m × 0.25 mm × 0.25 μm; Agilent Technologies, Santa Clara, CA, USA) was utilized. Each 1 μL sample was injected in the splitless mode. The olfactory port temperature was maintained at 250 °C. The oven temperature was initially set at 40 °C. It was then increased to 50 °C at a rate of 10 °C per minute and held for 20 min. Next, it was raised at 1 °C per minute to 70 °C and held for 10 min. Finally, it was increased at 3 °C per minute to 250 °C and held for 15 min. The identification of compounds was conducted in a full scan mode. The mass range was set from 45 to 350 amu.

In addition to standard comparison and the retention index (RI), a compound was identified if the similarity between the MS information of each chromatographic peak and the National Institute of Standards and Technology (NIST) or the LIQUOR (team self-built mass spectra library) was at least 80%.

### 2.5. Odor-Specific Magnitude Estimation (Osme)

During a GC–O–MS run, a panelist placed his/her nose near the olfactory port and recorded the aroma characteristic, intensity, and retention time of the chromatographic effluent. The aroma characteristics referred to previous references [26,27]. The aroma characteristics were familiarized by three panelists based on the provided standard. The aroma intensity judgment included 0 = none, 1 = very weak, 2 = weak, 3 = moderate, 4 = strong, and 5 = very strong. The Osme value for the aroma intensity was an average assessment performed by the panelists [28].

### 2.6. Quantification and Odor Active Values (OAVs)

The IS 1–3 were added to three samples at final concentrations of 40 mg/L, 20 mg/L, and 20 mg/L, respectively. The samples were then added into the GC with SPME in the splitless mode under the conditions described above. All the analyses were repeated three times. The odor thresholds referred to the previous research [29,30,31]. The OAVs of the flavor compounds were calculated according to the ratio of the target odorant concentration to their threshold.

### 2.7. Descriptive Sensory Analysis (DSA)

The study was reviewed and approved by the Beijing Technology and Business Univ. IRB, approval number 145 in 2024, and informed consent was obtained from each assessor (8 females and 8 males) prior to their participation in the study.

Over 30 students and employees were selected as panel candidates. The general training schedule spanned 8 weeks, with 2 h per week dedicated to introducing sensory analysis, aroma description and identification, ranking intensity, and conducting triangle tests. Sensory attributes were selected according to the aroma and taste characteristics of beer, including malty aroma, floral aroma, fruity aroma, honey aroma, nutty aroma, green aroma, smoky aroma, sweet taste, bitter taste, sour taste, spicy taste, greasy taste, carbon dioxide taste, freshness, and preference. The performances of candidates were recorded to ensure the effectiveness of the training. A total of 16 candidates (8 females and 8 males) with good performances were chosen as the DSA panel. The sensory attributes mentioned above were rated on a scale of 0 to 5 by the DSA panel.

### 2.8. Recombination and Omission Experiments

The odorants were dissolved in three ethanol solutions according to their detected concentrations in the three lager beer samples. The alcohol contents (*v*/*v*) of these ethanol solutions were 8%, which were the same as those of the original samples. The simulated samples were evaluated as described in Section 2.7, and compared with their corresponding original samples [32].

The omission tests were conducted to further confirm the key odorants in the lager beer. A comparative analysis of the ellipsis models and the respective completed recombination models was performed using the triangle tests. The significance levels of the detected difference were calculated according to the previous method [27].

### 2.9. Validation Experiments

The original beer was divided into 20 samples to which were added different concentrations of ethyl acetate (200 μg/L and 500 μg/L), isopentyl acetate (100 μg/L and 600 μg/L), ethyl hexanoate (300 μg/L and 400 μg/L), ethyl octanoate (600 μg/L and 1200 μg/L), phenethyl alcohol (2800 μg/L and 3300 μg/L), octanoic acid (20,000 μg/L and 31,000 μg/L), decanoic acid (5000 μg/L and 12,000 μg/L), γ-decanolactone (10 μg/L and 40 μg/L), and 2,4-bis(1,1-dimethylethyl)-phenol (5 μg/L and 15 μg/L). Each 35 mL was taken as the test sample and the INSENT taste analysis system (Beijing Ensoul Technology Ltd., Beijing, China) was used for taste detection. The INSENT taste analysis system mainly simulates the human taste perception mechanism through a multi-channel sensor array and can specifically respond to basic taste substances such as bitterness, sweetness, sourness, and so on. The aroma attributes of the samples were rated on a scale of 0 to 5 by the DSA panel.

The INSENT taste analysis system consisted of five taste sensors for sour, sweet, bitter, salty, and umami tastes, as well as two Ag/AgCl reference electrodes. Firstly, the sensor was cleaned in positive and negative cleaning solutions for 90 s, respectively, and then cleaned twice in the reference solution for 120 s each time. Then, the sensor was immersed in a reference solution and equilibrated to zero for 30 s. After passing the system self-test, the sample testing began. Each sample was tested for 30 s, followed by a 30 s aftertaste test. The sweetness analysis was repeated 5 times, and other taste analyses were repeated 4 times. The data from 3 times were taken as the test results, and the average value was calculated.

### 2.10. Statistical Analysis

Based on the results mentioned above, the dataset of sensory attributes of the samples and the dataset of compounds were created. The correlation between the two datasets was analyzed using the Pearson correlation coefficient. The data of taste in the validation experiments was analyzed by using the INSENT system. Statistical analyses were performed using the SPSS 22.0 (SPSS Inc., Chicago, IL, USA). Graphics were drawn using Origin 2021 (OriginLab Co., Northampton, MA, USA).

## 3. Results

### 3.1. Chemical Analysis

#### 3.1.1. Characterization and Expression Analysis of the Flavor Compounds

To facilitate the identification, SAFE combined with GC×GC-MS was applied to qualitatively analyze the trace components in three samples. As a result, lots of trace components were found in lager beer, among which alcohols and esters showed significant advantages, followed by acids. Generally, 617 trace components were identified in sample A, 700 trace components were detected in sample B, and 881 trace components were found in sample C. As shown in Figure 2, the number of esters was the highest in sample C, and the number of acids in sample A was the highest. The number of alcohols and ketones was also relatively high in sample C. Moreover, sulfur and nitrogen compounds, as important trace components in beer, had little difference in their number between samples A and B, but were the most abundant in sample C. However, not all the trace components had a significant impact on the flavor characteristics of beer; only a portion of them contributed significantly to the flavor profile of lager beer.

Therefore, Osme combined with SPME, LLE, and GC–O–MS was applied to further analyze the flavor compounds in three samples. The results indicated that alcohols and esters primarily have fruity and sweet aromas, whereas acids showed a stimulating aroma. Aromatic compounds were mostly identified as nut and plant aromas. Moreover, the aroma intensity of alcohols and acids was the highest in the three samples, followed by esters and aromatic compounds. The aroma intensity of the same flavor compound also varied in the three samples. For example, great differences in the intensity of alcohols and acids were found among the three samples. Specifically, 3-methyl-1-butanol had the highest aroma intensity in sample A, while decanoic acid showed the highest aroma intensity in sample B. The aroma intensity of hexanoic acid was higher in sample C. It was indicated that the combined effect of these flavor compounds with different characteristics might affect the overall flavor profile of lager beer. In addition, the aroma intensity of flavor compounds might vary under different pre-treatment conditions as shown in Table 1, which was speculated to be related to the ability of different pretreatment methods to enrich and extract the flavor compounds. Hence, a combination of multiple pre-treatment methods might be recommended for future research on the flavor compounds in beer.

#### 3.1.2. Quantification and OAVs of the Flavor Compounds

Based on the results above, the flavor compounds were further quantitatively analyzed via GC-MS/SIM. The results showed that the concentrations of octanoic acid, decanoic acid, hexanoic acid, phenethyl acetate, and phenylethyl alcohol were relatively high in lager beer. Specifically, the concentration of hexanoic acid was the highest in sample A, the concentrations of phenylethyl acetate and phenylethyl alcohol were the highest in sample B, and the concentrations of octanoic acid and decanoic acid were the highest in sample C. According to the results of Section 3.1.1, a certain correlation between the aroma intensity and concentrations of flavor compounds were found in lager beer. In addition, some flavor compounds were only detected in individual samples, such as butyl butyrate and nonanoic acid, which were only found in sample B. The different concentration was considered to be a key factor influencing the expression of the flavor compounds, which might affect the flavor profiles of the samples.

The flavor profile of beer was influenced not only by the concentrations of the flavor compounds but also by their interactions and the matrix effect. Previous studies have shown that ethanol might affect the perception of these compounds. OAVs indicate the ratio of the flavor compounds concentration to the odor threshold in the relevant medium. As shown in Table 2, the OAVs of these flavor compounds were calculated based on the quantitative results mentioned above. According to the results, a total of seven flavor compounds with OAVs ≥ 1 were confirmed in the three samples, which were isopentyl acetate, ethyl octanoate, phenylethyl acetate, ethyl hexanoate, hexanoic acid, octanoic acid, and 4-octanone. Although its concentration was not the highest, the OAV of ethyl octanoate was relatively high due to its low threshold, which made an important contribution to the flavor characteristics of lager beer. The OAV of ethyl octanoate varied among the three samples, with the highest OAV calculated in sample C, which was consistent with the results in Section 3.1.1. It was preliminarily speculated that ethyl octanoate had a greater effect on the flavor profile of sample C. On the contrary, the OAVs of some flavor compounds with a higher concentration were not high due to the influence of the odor threshold, such as hexanoic acid, decanoic acid, and phenethyl acetate, etc. Further verification was still needed to analyze how these flavor compounds with a strong aroma expression contributed to the overall flavor profile of lager beer.

### 3.2. Sensory Evaluation

#### 3.2.1. Overall Sensory Evaluation

The overall sensory evaluation of three lager beers was carried out, in which 15 attributes (malty aroma, floral aroma, fruity aroma, honey aroma, nutty aroma, green aroma, smoky aroma, sweet taste, bitter taste, sour taste, spicy taste, greasy taste, carbon dioxide taste, freshness, preference) of the three samples were also scored 0–5 by the DSA panel. The attributes were determined based on the beer flavor wheel and the evaluation results of the DSA panel [33]. As shown in Figure 3a, the flavor profile of lager beer included clear fruity aroma, honey aroma, green aroma, sweet taste, sour taste, and freshness. The smoky aroma, spicy taste, and greasy taste were lighter, while the malty and floral aromas were relatively moderate. Hence, this overall flavor profile made lager beer more suitable for the daily lives of consumers, especially on hot days or when eating fatty food. In addition, some dissimilarities of the same flavor characteristics were also found in the three samples. Specifically, sample C had the highest intensities of honey aroma and carbon dioxide taste, which had significant differences from samples A and B. Meanwhile, the intensity of freshness was the strongest in sample B, and the highest intensities of preference were found in sample A, which had significant differences among the three samples. These minor dissimilarities in the flavor characteristics of lagers were also considered to enrich the choices of consumers in their daily lives. The differential expression of the flavor compounds in the samples was speculated to be one of the factors affecting the above results. To further study the importance of these compounds in relation to the flavor profiles of the three lager beers, the recombination experiments and omission experiments were conducted according to the results of Section 3.1.

#### 3.2.2. Recombination Experiments

According to the results mentioned above, the flavor compounds were mixed to prepare a model solution that matched their concentrations in the original samples (Table 2). The seven flavor attributes (malty aroma, floral aroma, fruity aroma, honey aroma, nutty aroma, green aroma, smoky aroma) of the three samples and their simulated model solutions were evaluated by the DSA panel. As shown in Figure 3b–d, the flavor profile of the simulated model closely matched that of the original samples (Pearson coefficients ≥ 0.9). This indicated that the primary flavor profile of lager beer is related to the combined action of these flavor compounds and the changes in these flavor compounds might affect the flavor profile of lager beer. However, some intensities of flavor characteristics in the recombinant samples were still inconsistent with those in the original samples. Specifically, the intensity of smoky aroma in the recombinant samples was generally higher than that in the original samples, while the intensities of malty aroma and green aroma in the recombinant samples were slightly weaker than those in the original samples. It was also found that there was a difference in the intensity of nutty aroma between the recombinant sample A and the original sample A. The intensity of honey aroma in the recombinant sample B was not as strong as that in the original sample B. Meanwhile, compared with the recombinant sample C, the intensities of floral aroma and honey aroma were higher in the original sample C. It could be inferred that some trace compounds might also have a slight effect on these sensory attributes except the aforementioned flavor compounds. Furthermore, the influences of important flavor compounds on the flavor profile were confirmed via omission experiments and a flavor matrix.

#### 3.2.3. Omission Experiments

The omission experiments were conducted to further analyze the contributions of important flavor compounds to the overall flavor profile of lager beer. Triangle tests were carried out by the DSA panel, and one or a set of flavor compounds with the same functional group were omitted to obtain the model solutions. As shown in Table 3, model 1, model 2, and model 24 indicated that most of the assessors could identify the absence of esters (*p* ≤ 0.001), indicating that ethyl acetate, isopentyl acetate, and all acids were critical to the flavor profile of lager beer. This was similar to previous research results. In 2024, the research also found that ethyl acetate and isoamyl acetate were the key flavor compounds in beer, which could have an impact on the flavor profile of beer [34]. However, the significance values in the three samples were not all high when all the esters were omitted, which might be related to the interaction between the flavor compounds.

Furthermore, most of the assessors could recognize the omission of all alcohols in samples A and B, while samples with missing individual alcohols, such as 3-methyl-1-butanol and 1-octanol, could not be distinguished well in the samples. According to the results of the omission experiments, although all the alcohols made a significant contribution to the flavor profile of lager beer, not all the individual alcohol compounds did so. Meanwhile, phenethyl acetate and furfural were calculated to have high OAVs (Table 3), but few of the assessors could recognize the omission of them in the three samples. It was preliminary speculated that their impact on the overall flavor profile of lager beer was limited. In addition, the same flavor compound also showed different significances in the three samples, such as ethyl hexanoate and octanoic acid, and was similar to the results mentioned in Section 3.1. This might be one of the causes of the differences in the flavor characteristics among the three samples. Generally, ethyl acetate, isopentyl acetate, and all the acids were speculated to have an impact on the overall flavor profile of lager beer according to the omission test. The different expressions of these compounds might affect the overall flavor profile of lager beer. However, further research is needed to explore the relationship between flavor compounds and sensory attributes in lager beer.

#### 3.2.4. Correlation Analysis of Flavor Compounds and Flavor Profile of Lager Beer by Using a Flavor Matrix

Based on the results above, this study further explored the flavor profile and preference of lager beer. Except aroma attributes, the six taste attributes and two evaluation attributes (sweet taste, bitter taste, sour taste, spicy taste, greasy taste, carbon dioxide taste, freshness, preference) of the three samples were also scored by the DSA panel. The Pearson correlation coefficient was calculated and plotted to show the relationship between the concentrations of flavor compounds and sensory attributes. As shown in Figure 4, the orange bubbles represent a positive correlation between the flavor compounds and attributes, while the blue ones represent a negative correlation. Meanwhile, the size of the bubbles was used to indicate the magnitude of the correlation value. According to Section 3.2.2, lager beer has distinct characteristics of a floral aroma, fruity aroma, and honey aroma. As shown in Figure 4, floral aroma, fruity aroma, and honey aroma were positively correlated with most esters and alcohols such as ethyl acetate, ethyl octanoate, and 1-octanol. Octanoic acid and decanoic acid made a positive contribution to the malty aroma, floral aroma, fruity aroma, honey aroma, and spicy taste of beer, similar to some esters. Moreover, these two acids also had a significant positive impact on the nutty aroma, smoky aroma, bitter taste, sour taste, and carbon dioxide taste. It was speculated that this might be related to the high quantity and concentration of these flavor compounds.

Lager beer is widely welcomed by consumers due to its light and refreshing flavor characteristics. As shown in Figure 4, the flavor compounds that had a significant positive correlation with the freshness of samples mainly included phenylethyl acetate, ethyl 3-phenylpropionate, 2-furanmethanol, and dodecanoic acid. A correlation analysis between the preference for lager beer and flavor compounds was also conducted. It was found that ethyl 9-decanoate, hexanoic acid, and dodecanoic acid had a strong positive impact on the preference. On the contrary, ethyl acetate, ethyl propionate, ethyl octanoate, and decanoic acid showed a negative correlation with the preference, which might be related to the positive effects of these flavor compounds on the spicy taste.

Surprisingly, some esters in the samples exhibited a negative correlation with a fruity aroma, such as isopentyl acetate, which was different from their individual aroma characteristics. This might be due to the interaction of multiple flavor compounds in beer. This was also similar to previous research. For example, in 2018, a negative correlation between ethyl dodecanoate and the floral aroma of beer was found in beers brewed with different malts [18]. Generally, the correlation between different compounds and the flavor characteristics of samples varied, and the combined effect of these flavor compounds contributed to form the unique flavor profile of lager beer. The influence of the same compounds on the flavor characteristics at different concentrations still needs further verification.

#### 3.2.5. The Validation Experiment for the Flavor Compounds

Owing to the relative results of the quantitative and sensory experiments mentioned above, nine flavor compounds were identified as important flavor compounds in lager beer including ethyl acetate, isopentyl acetate, ethyl hexanoate, ethyl octanoate, phenylethyl alcohol, octanoic acid, decanoic acid, γ-decalactone, and 2,4-bis(1,1-dimethylethyl)-phenol. In order to further verify the contribution of the important flavor compounds at different concentrations to the taste of lager beer, an electronic tongue was applied to detect the sourness, sweetness, bitterness, astringency, richness, saltiness, and aftertaste of lager and 20 samples with the addition of nine compounds. The results showed that the addition of nine important flavor compounds at different concentrations all had an impact on the sourness of lager beer. Specifically, esters, alcohols, and acids all had positive effects on the sourness at low concentrations, while they made negative contributions to the sourness at high concentrations. This indicated that adjusting the concentration of esters, alcohols, and acids could have a certain impact on the sourness of lager beer.

The low concentrations of 2,4-bis(1,1-dimethylethyl)-phenol could weaken the sweetness of lager beer, which was similar to the result in Figure 4. While nine flavor compounds with high concentrations all had a significant positive effect on the sweetness of beer, bitterness was also considered one of the essential factors which affect the flavor profile of beer. As shown in Figure 5, it was found that the negative effects of aromatic compounds on the bitterness increased significantly with the increase in concentrations, while esters and alcohols had more negative effects on the bitterness at low concentrations. Notably, acids weakened the bitterness of beer at low concentrations and strengthened it at high concentrations.

In terms of the mouthfeel of the samples, the effects of flavor compounds on the astringency were relatively light, with alcohols and acids having a slightly negative effect on the astringency. It was also found that the addition of esters, alcohols, and acids would affect the richness of beer, and the impact on the richness decreased with the increase in concentrations. Moreover, the addition of all the flavor compounds had negative effects on the aftertaste. Among them, the low concentrations of aromatic compounds had the least impact on the aftertaste, while at high concentrations, octanoic acid, decanoic acid, and phenethyl alcohol had the least effect on the aftertaste.

In addition to the taste and mouthfeel, the aroma was also a key factor affecting the choice of lager by consumers. Hence, the aroma of the added and original samples was also verified by the DSA panel. As shown in Figure 5, the results implied that the more acids added, the greater the positive effects on the smoky aroma. It was also found that phenethyl alcohol, decanoic acid, and γ-decalactone had negative effects on the malty aroma at high concentrations. The effects of nine compounds on the floral aroma of beer were relatively insignificant at low concentrations, while these compounds except for decanoic acid had significant positive effects on the floral aroma at high concentrations. Meanwhile, the positive effects of ethyl acetate and isopentyl acetate on the fruity aroma increased with the increase in their concentrations, while octanoic acid showed a negative effect on the fruity aroma regardless of its concentration. Other esters and alcohols had negative effects on the fruity aroma at low concentrations, and they had significant positive effects on the fruity aroma at high concentrations. It was also found that ethyl hexanoate had a positive effect on the nutty aroma at low concentrations, but showed a negative effect at high concentrations, while phenethyl alcohol had a negative effect on the nutty aroma regardless of the concentration. Overall, the influence of the same compound on the same flavor characteristics might be completely opposite at different concentrations. It was preliminarily speculated that the flavor profile of lager beer mainly depended on the combination of flavor compounds within a certain concentration range.

## 4. Conclusions and Discussion

Several extraction methods combined with GC×GC-MS, GC-MS, and GC–O–MS were applied for the qualitative analysis of the trace components in lager beer. Then, 35 flavor compounds were further quantified, and their OAVs were also calculated. The flavor profile of lager beer was successfully simulated, and the results of the omission test preliminarily showed that ethyl acetate, isopentyl acetate, phenylethyl alcohol, and decanoic acid had a significant impact on the flavor profile of lager beer. Then, the relationship between the flavor compounds and 15 attributes was calculated and illustrated. A total of nine flavor compounds including ethyl acetate, isopentyl acetate, ethyl hexanoate, ethyl octanoate, phenylethyl alcohol, octanoic acid, decanoic acid, γ-nonanolactone, and 2,4-bis(1,1-dimethylethyl)-phenol were identified as important flavor compounds in lager beer. Moreover, a validation experiment was conducted to further verify their effects on the flavor profile of lager beer at different concentrations. Consequently, various flavor compounds were found with different degrees of influence on the overall flavor profile, while even the same flavor compound might have opposite effects on the same flavor characteristics at different concentrations. Hence, the overall flavor profile of lager beer might be formed by these important flavor compounds with different flavor expressions at various concentrations. In the future, due to technological advancements and the pursuit of high-quality living, the demand for various types of alcoholic beverages will continue to increase, and lager beer has always been an important part of this. These results provide a targeted basis for measuring and improving the quality of lager beer, and we will further study the related problems in the future research.

## Figures and Tables

**Figure 1 foods-14-01702-f001:**
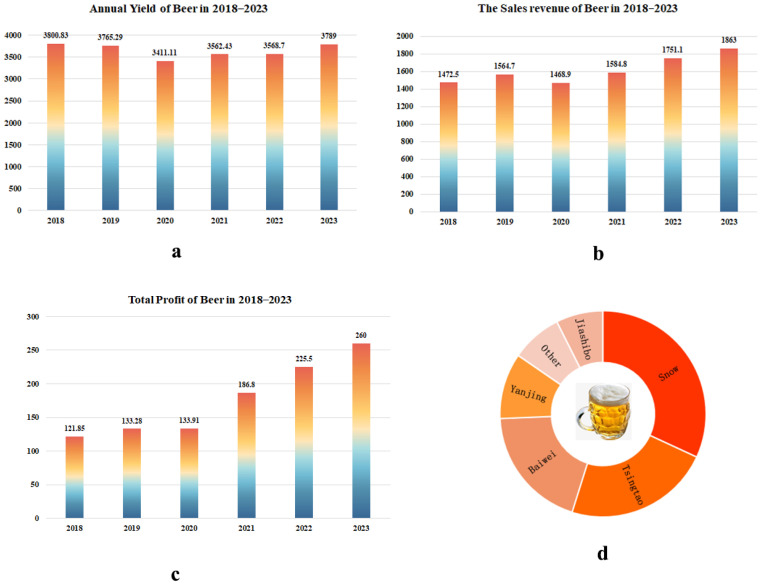
Overview of China beer industry in the period 2018–2023. (**a**) Annual yield of beer in the period 2018–2023. (**b**) The sales revenue of beer in the period 2018–2023. (**c**) Total profit from beer in the period 2018–2023. (**d**) Main brands in the Chinese beer market in 2023.

**Figure 2 foods-14-01702-f002:**
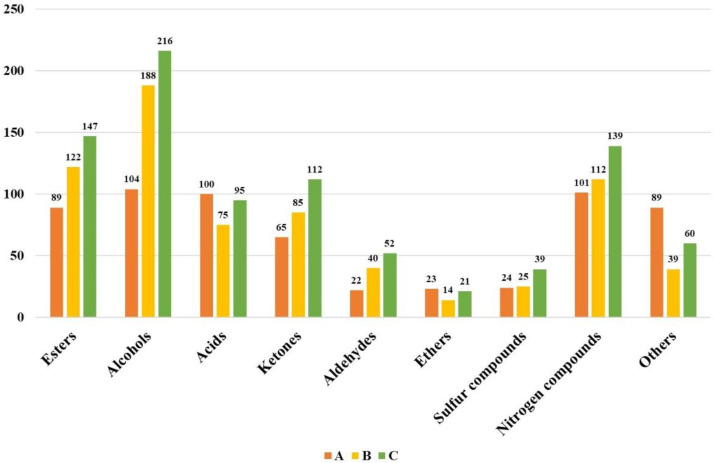
Qualitative results of trace components in three lager beers via GC×GC-MS.

**Figure 3 foods-14-01702-f003:**
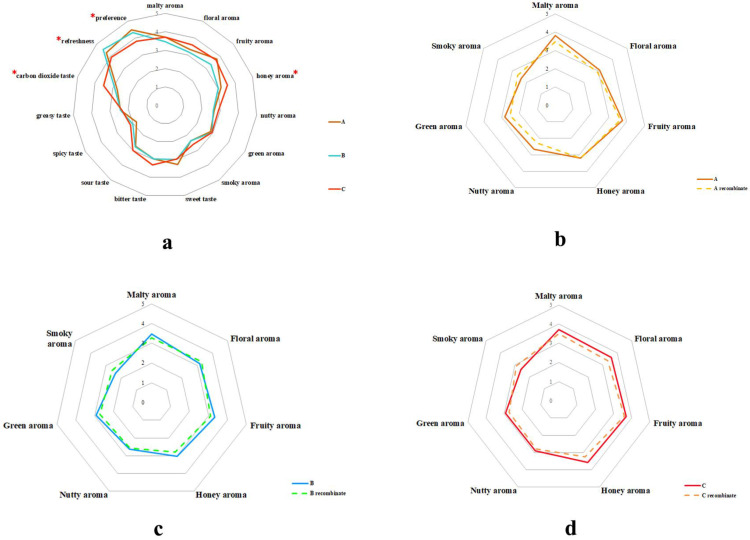
Sensory evaluation of three lagers. (**a**) The overall flavor profile of three lager beers. *: there were significant differences among three samples (*p* < 0.05). (**b**–**d**) Aroma recombination of three lagers, (**b**) sample A, (**c**) sample B, (**d**) sample C.

**Figure 4 foods-14-01702-f004:**
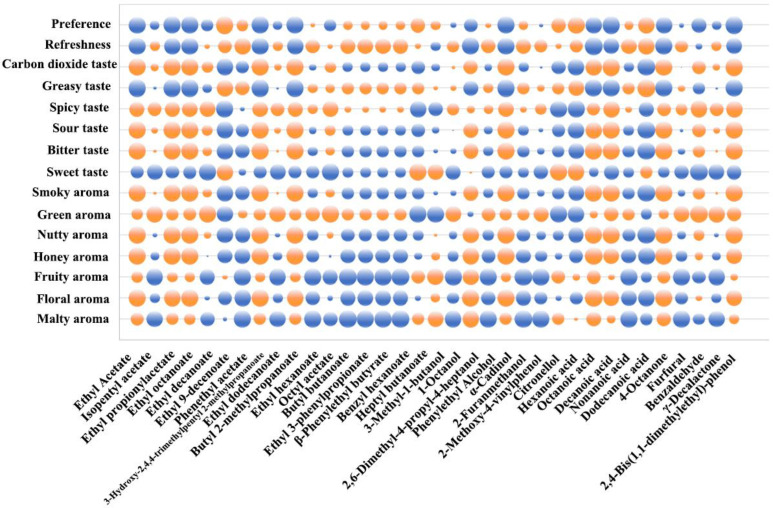
The correlations between flavor compounds and flavor characteristics. The orange bubbles: positive correlations; the blue bubbles: negative correlations; the size of the bubbles was used to indicate the magnitude of the correlation value.

**Figure 5 foods-14-01702-f005:**
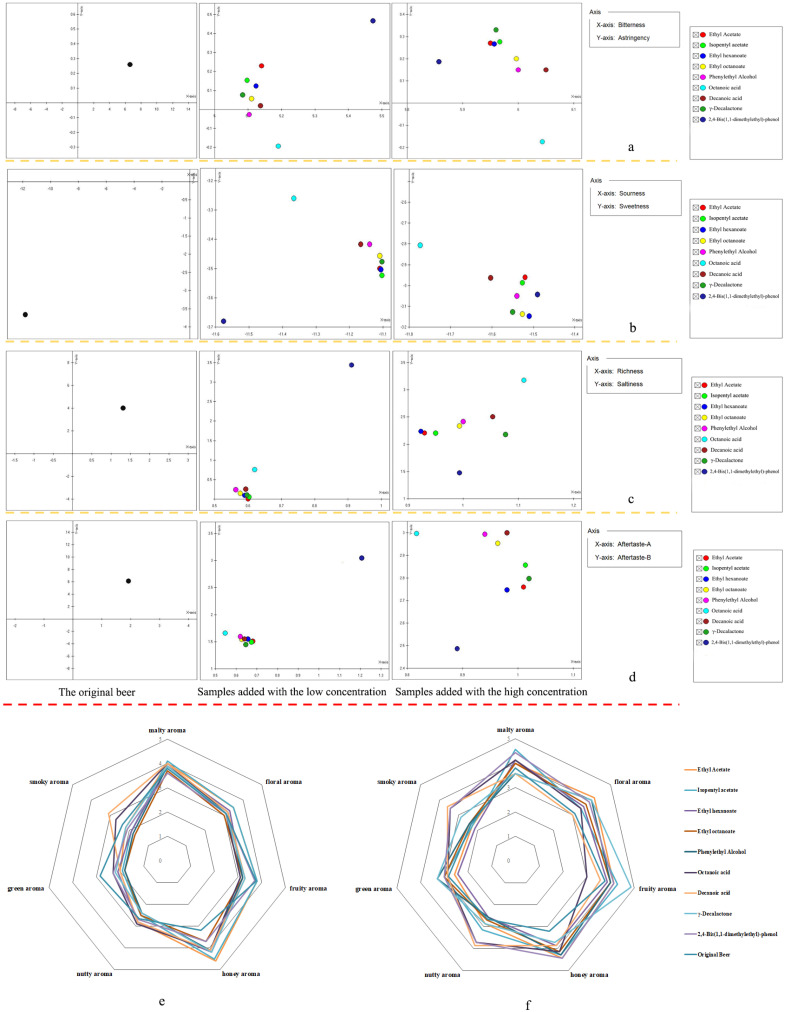
The results of validation experiment. (**a**–**d**) The results of the taste characteristics, from left to right, were the original beer, low addition group, and high addition group, respectively. (**e**) The results of aroma characteristics in the low addition group. (**f**) The results of aroma characteristics in the high addition group.

**Table 1 foods-14-01702-t001:** The aroma intensity and characteristics of flavor compounds.

Flavor Compounds	CAS	Aroma Characteristics	Aroma Intensity											
			A				B				C			
			SPME	SD	LLE	SD	SPME	SD	LLE	SD	SPME	SD	LLE	SD
Ethyl Acetate	141-78-6	Fruity			1	0.24			1	0.24	2	0.24	2	0.24
Isopentyl acetate	123-92-2	Banana					1	0.24	3	0.47	2	0.24	3	0.24
Ethyl propionylacetate	4949-44-4	Sweet									1	0.24	1	0.24
Ethyl hexanoate	123-66-0	Pineapple, Peach	1	0.24	2	0.24	2	0.24	2	0.24	2	0.24	1	0.24
Ethyl octanoate	106-32-1	Pear	1	0.24	1	0.24	1	0.24	1	0.24	3	0.41	2	0.24
Octyl acetate	112-14-1	Mushroom									2	0.24	1	0.24
Ethyl decanoate	110-38-3	Fruity, Peach	2	0.24			2	0.41	1	0.24	1	0.24	1	0.24
Ethyl 9-decenoate	67233-91-4	Fruity, Fatty			1	0.24	1	0.24	1	0.24				
2-Phenethylformate	104-62-1	Green, Floral	2	0.24										
Phenethyl acetate	103-45-7	Rose, Honey	4	0.24	4	0.41	3	0.24	3	0.24	4	0.41	4	0.47
Ethylene glycol	107-21-1	Sweet							3	0.41				
2-Furanmethanol	98-00-0	Bready							1	0.24				
1-Octanol	111-87-5	Coconut									1	0.24	2	0.24
3-Methyl-1-butanol	123-51-3	Sugarcane, Fruity	4	0.24	2	0.24	3	0.41	2	0.24	1	0.24	1	0.24
2,3-Butanediol	513-85-9	Buttery, Creamy			1	0.24			2	0.24	2	0.24	1	0.24
3-Methylthiopropanol	505-10-2	Vegetable			2	0.24			1	0.24	1	0.24	1	0.24
Phenylethyl Alcohol	60-12-8	Floral	4	0.47	5	0.41	3	0.41	4	0.24	5	0.47	4	0.24
2-Ethylbutyric acid	88-09-5	Dairy	2	0.24	2	0.24	2	0.24	1	0.24	2	0.24	3	0.24
2-(Aminooxy)valeric acid	5699-55-8	Earthy	1	0.24										
Hexanoic acid	142-62-1	Cheesy, Fruity	1	0.24	3	0.24	1	0.24	1	0.24	3	0.24	3	0.24
2-Methyl-2-pentenoic acid	3142-72-1	Jammy							1	0.24				
Octanoic acid	124-07-2	Cheesy	2	0.24	1	0.24	3	0.24	2	0.24	1	0.24	3	0.24
Decanoic acid	334-48-5	Fatty	2	0.24			4	0.24	3	0.24	1	0.24	3	0.41
Benzaldehyde	100-52-7	Nutty, Fruity	2	0.41	1	0.24	1	0.24	1	0.24			1	0.24
Cinnamaldehyde	104-55-2	Honey, Sweet	1	0.41			2	0.24	1	0.24				
Furfural	98-01-1	Woody, Nutty	1	0.24							2	0.24	1	0.24
Naphthalene	91-20-3	Fatty	3	0.47	3	0.24								
Benzothiazole	95-16-9	Vegetable	3	0.41										
4-Hydroxy-2,5-Furaneol	3658-77-3	Caramel			3	0.47			2	0.24			2	0.24
Phenol	108-95-2	Rubbery	2	0.24	1	0.24	2	0.24	1	0.24				
2-Acetyl pyrrole	1072-83-9	Nutty											2	0.41
γ-Nonanolactone	104-61-0	Floral	3	0.47	4	0.24	2	0.24	1	0.24	3	0.24	3	0.24
2-Methoxy-4-vinylphenol	7786-61-0	Smoky	1	0.24	3	0.24	1	0.24	2	0.24				

The values in the table were the average results, and the standard deviation (SD) was calculated using the scores of three panelists which ranged from 0 to 5 in increments of 0.5.

**Table 2 foods-14-01702-t002:** The concentrations and OAVs of flavor compounds in lager beer.

Flavor Compounds	CAS	Odor Threshold (μg/L)	Concentration (μg/L)			OAV		
			A	B	C	A	B	C
Ethyl Acetate	141-78-6	32,552 ^b^	262.81 ± 14.32	250.88 ± 19.60	457.99 ± 14.00	0	0	0
Isopentyl acetate	123-92-2	94 ^b^	164.21 ± 3.20	595.02 ± 14.25	394.83 ± 13.02	2	6	4
Ethyl propionylacetate	4949-44-4		0	0	13.70 ± 4.14			
Ethyl octanoate	106-32-1	13 ^b^	640.7 ± 10.52	657.43 ± 9.43	1358.35 ± 11.71	49	51	104
Ethyl decanoate	110-38-3	1120 ^b^	83.40 ± 1.71	228.26 ± 7.92	194.20 ± 7.27	0	0	0
Ethyl 9-decenoate	67233-91-4		24.16 ± 4.01	21.77 ± 1.80	18.79 ± 2.90			
Phenethyl acetate	103-45-7	407 ^b^	2113.72 ± 18.82	2782.11 ± 19.07	1910.71 ± 5.49	5	7	5
3-Hydroxy-2,4,4-trimethylpentyl 2-methylpropanoate	74367-34-3		0	0	18.05 ± 4.40			
Ethyl dodecanoate	106-33-2	500 ^b^	0	26.81 ± 5.34	14.11 ± 5.84		0	0
Butyl 2-methylpropanoate	97-87-0	22,000 ^a^	0	0	24.65 ± 6.96			0
Ethyl hexanoate	123-66-0	55 ^b^	317.02 ± 9.67	362.94 ± 4.93	326.62 ± 14	6	7	6
Octyl acetate	112-14-1	450 ^d^	0	11.31 ± 4.92	8.52 ± 0.48		0	0
Butyl butanoate	109-21-7	110 ^b^	0	27.47 ± 3.54	0		0	
Ethyl 3-phenylpropionate	2021-28-5	125 ^b^	0	19.20 ± 1.35	0		0	
β-Phenylethyl butyrate	103-52-6		0	10.64 ± 1.81	0			
Benzyl hexanoate	6938-45-0		0	10.67 ± 2.98	0			
Heptyl butanoate	5870-93-9		18.24 ± 3.63	0	0			
3-Methyl-1-butanol	123-51-3	10,000 ^c^	302.95 ± 2.43	249.06 ± 5.35	272.20 ± 13.27	0	0	0
1-Octanol	111-87-5	1100 ^c^	20.10 ± 1.01	26.28 ± 1.95	22.55 ± 0.33	0	0	0
2,6-Dimethyl-4-propyl-4-heptanol	54774-83-3		22.50 ± 1.98	0	36.96 ± 1.69			
Phenylethyl Alcohol	60-12-8	47,900 ^a^	2854.20 ± 14.93	3255.28 ± 15.28	2921.54 ± 8.18	0	0	0
α-Cadinol	481-34-5		0	0	7.56 ± 0.31			
2-Furanmethanol	98-00-0	1400 ^c^	0	7.00 ± 1.24	0		0	
2-Methoxy-4-vinylphenol	7786-61-0	500 ^a^	0	44.04 ± 4.78	14.66 ± 6.27		0	0
Citronellol	106-22-9	700 ^c^	8.03 ± 2.13	0	0	0		
Hexanoic acid	142-62-1	2517 ^b^	1415.51 ± 10.11	1362.31 ± 18.46	1317.68 ± 11.32	1	1	1
Octanoic acid	124-07-2	2701 ^b^	25,627.31 ± 12.53	23,553.12 ± 18.06	30,402.23 ± 10.11	9	9	11
Decanoic acid	334-48-5	13,737 ^b^	5277.46 ± 17.35	6660.27 ± 14.87	11,151.89 ± 19.68	0	0	1
Nonanoic acid	112-05-0	2400 ^c^	0	365.00 ± 11.93	0		0	
Dodecanoic acid	143-07-7	10,000 ^c^	143.30 ± 18.22	154.69 ± 7.85	0	0	0	
4-Octanone	589-63-9	82 ^d^	124.09 ± 9.29	122.09 ± 5.08	132.24 ± 7.30	2	1	2
Furfural	98-01-1	15,000 ^a^	81.06 ± 7.85	115.50 ± 10.00	93.21 ± 17.46	0	0	0
Benzaldehyde	100-52-7	5000 ^c^	0	25.94 ± 3.46	26.44 ± 1.89		0	0
γ-Decalactone	706-14-9	1000 ^c^	15.02 ± 1.71	32.58 ± 1.37	24.31 ± 5.80	0	0	0
2,4-Bis(1,1-dimethylethyl)-phenol	96-76-4	500 ^d^	9.67 ± 0.69	10.56 ± 1.22	13.90 ± 1.13	0	0	0

Matrix solution for odor threshold: ^a^—beer; ^b^—baijiu; ^c^—ethanol aqueous solution; ^d^—water.

**Table 3 foods-14-01702-t003:** Omission experiments from complete reconstitution of flavor compounds.

NO.	Flavor Compounds Omitted	*n*			Significance		
		A	B	C	A	B	C
1	Ethyl acetate	14	13	14	***	***	***
2	Isopentyl acetate	12	12	13	***	***	***
3	Ethyl hexanoate	11	6	14	**		***
4	Ethyl propionylacetate	——	——	11			**
5	Ethyl octanoate	——	11	10		**	*
6	Octyl acetate	——	11	11		**	**
7	Ethyl decanoate	10	9	11	*	*	**
8	Phenethyl acetate	10	4	11	*		**
9	Ethyl dodecanoate	——	10	13		*	***
10	Butyl butanoate	——	8	14			***
11	β-Phenylethyl butyrate	——	9	——		*	
12	All esters	12	11	11	***	**	**
13	3-Methyl-1-butanol	9	9	7		*	
14	Citronellol	10	——	——	*		
15	Phenylethyl alcohol	13	14	11	***	***	**
16	1-Octanol	——	12	10		***	*
17	2-Furanmethanol	——	15	——		***	
18	All alcohols	12	14	11	***	***	**
19	Hexanoic acid	——	14	11		***	**
20	Octanoic acid	11	13	10	**	***	*
21	Nonanoic acid	——	10	——		*	
22	Decanoic acid	11	14	14	**	***	***
23	Dodecanoic acid	11	9	——	**	*	
24	All acids	14	12	13	***	***	***
25	Furfural	10	9	6	*	*	
26	γ-Decalactone	11	14	——	**	***	
27	2,4-Bis(1,1-dimethylethyl)-phenol	11	11	11	**	**	**

*n*: number of correct judgments from 16 assessors evaluating the aroma difference by using the triangle test; ——: This compound was not included in the sample; significance: *, significant (*p* ≤ 0.05); **, highly significant (*p* ≤ 0.01); ***, very highly significant (*p* ≤ 0.001).

## Data Availability

The original contributions presented in the study are included in the article, further inquiries can be directed to the corresponding author.

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
