# Peer review of "Characterization of Key Factors Associated with Flavor Characteristics in Lager Beer Based on Flavor Matrix"

_foods, 2025, doi:10.3390/foods14101702_

Round 1
Reviewer 1 Report
Comments and Suggestions for Authors
Dear Authors,
The topic of your manuscript, entitled "Characterization of key factors associated with flavor characteristics in lager based on flavor matrix" is very interesting and a lot of experimental work was done. However, there are some things that should be clarified in order to improve manuscript quality.
- Introduction -> There is no need of Figure 1. ln.73 AEDA and OAV are mentioned for first time in the manuscript, so please, write their full names
- Materials and methods ->ln. 96 Raw materials in beer production always include yeast. Beer ingridients don't include yeast because in commercial beers yeast are removed by centrufigation and filtration. Therefore, they are not written on the label. So, change raw materials to ingridients
- Results->ln. 235 As a whole only fatty acids showed cheese and fatty aroma
- Conclusion -> It will be good to have a conclusion
- General comments -> The manuscript is interesting but there are some thing that bother me. First, when you make model solutions you use only ethanol and water, without carbon dioxide or hop. So, I don't think that it corresponds to beer composition and the comparison is not correct. Second, it's strange that you can't find vicinal diketones or acetaldehyde in beer because they are formed during beer fermentation and affect negatively beer flavour.
Author Response
Comment 1: The topic of your manuscript, entitled "Characterization of key factors associated with flavor characteristics in lager based on flavor matrix" is very interesting and a lot of experimental work was done. However, there are some things that should be clarified in order to improve manuscript quality.
Response: It’s our pleasure to invite you to review this paper and thank you very much for your encouraging advices and careful review concerning our manuscript. We will carefully revise the text according to your guidance.
Comment 2: Introduction -> There is no need of Figure 1. ln.73 AEDA and OAV are mentioned for first time in the manuscript, so please, write their full names.
Response: Thank you for your patient reading and kind advices. The changed parts were marked by red in our revised manuscript. We have added their full names according to your advices.
Comment 3: Materials and methods ->ln. 96 Raw materials in beer production always include yeast. Beer ingridients don't include yeast because in commercial beers yeast are removed by centrufigation and filtration. Therefore, they are not written on the label. So, change raw materials to ingridients.
Response: Thank you for your patient reading and kind advices. The revised parts were marked in red in our revised manuscript.
Comment 4: Results->ln. 235 As a whole only fatty acids showed cheese and fatty aroma
Response: Many thanks for your patient reading and kind advices. The revised parts were marked by red in our revised manuscript. We have revised it into stimulating aroma.
Comment 5: Conclusion -> It will be good to have a conclusion.
Response: Thank you for kind guidance. The changed parts were marked by red in our revised manuscript. We have revised the part 4 according to your advice.
Comment 6: The manuscript is interesting but there are some thing that bother me. First, when you make model solutions you use only ethanol and water, without carbon dioxide or hop. So, I don't think that it corresponds to beer composition and the comparison is not correct. Second, it's strange that you can't find vicinal diketones or acetaldehyde in beer because they are formed during beer fermentation and affect negatively beer flavour.
Response: Thank you very much for your patient review and encouraging advices. We thought that alcohol content was one of the most important factors affecting flavor in alcoholic beverages, thus it was used as the primary medium for simulation experiments. Although the matrix solution was not identical to actual beer, we validated the significant contribution of these compounds to the beer through verification experiments, and the results were relatively reliable. We will adopt a more rigorous approach in the future experiments according to your suggestions, to ensure that the simulated solution closely resembles actual beer. In this study, we mainly analyzed the key flavor compounds. We thought vicinal diketones or acetaldehyde might volatilize and reduce more during long-term fermentation and filling, and their relative content were not relatively high, which might not contribute so much to the whole flavor. What you said was correct and we will improve experiments in the future according to your kind suggestions. Thank you again for your patient review.
Reviewer 2 Report
Comments and Suggestions for Authors
The manuscript offers valuable insights into the relationship between important flavor compounds and the flavor profile of lager beer. Although the manuscript contains valuable results, certain details regarding the analytical methods and result interpretation must be clarified.
Line 40-49: Each factual claim should be accompanied by a reference. Please revise accordingly.
Line 73: All abbreviations should be defined the first time they are mentioned.
Line 94: Please include the alcohol content (ABV) of the beer samples. If available, other basic characteristics such as bitterness (IBU), original gravity, or color would also be useful.
Line 166: It might be helpful to include a brief explanation of how the OAVs of the flavor compounds were calculated, especially for clarity and completeness.
Line 179: Please include the standard or guideline that was followed for sensory analysis.
Lines 190-198: Could you please clarify how the concentrations of the specific compounds to be used in the validation experiments were determined?
Line 199: Please provide a more detailed explanation of the INSENT taste analysis system used in the study.
Table 1: Since the aroma intensity is represented by the average result of panelists, the values in the table should be given as the average score and standard deviation. The table cannot contain empty cells
Table 2: Please clarify why the OAV values in Table 2 are presented as whole numbers, as this is not necessary. We encourage you to search for and include odor threshold values for all (or as many as possible) compounds. These values can also be found in studies related to wine, spirits, or other matrices (not necessarily related to beer). Including this information would enhance the discussion and improve the overall quality of the manuscript.
Figure 3 should be improved in terms of clarity and presentation. Additionally, since you had 16 sensory panelists, it is possible to perform statistical comparisons for each evaluated parameter. Please consider including indicators of significant differences among samples in the figure. This would strengthen the interpretation of the sensory data.
Author Response
Comment 1: The manuscript offers valuable insights into the relationship between important flavor compounds and the flavor profile of lager beer. Although the manuscript contains valuable results, certain details regarding the analytical methods and result interpretation must be clarified.
Response: We are very grateful to your comments for the manuscript. Thank you for your patient review and high evaluation of our work. We will revise it with red carefully according to your suggestions.
Comment 2: Line 40-49: Each factual claim should be accompanied by a reference. Please revise accordingly.
Response: Many thanks for your patient guidance. The data was obtained from the National Bureau of Statistics of China. We have added the website and the revised parts were marked in red in our revised manuscript.
Comment 3: Line 73: All abbreviations should be defined the first time they are mentioned.
Response: Thank you for your patient reading and kind advices. I agree with you and the changed parts were marked in red in our revised manuscript.
Comment 4: Line 94: Please include the alcohol content (ABV) of the beer samples. If available, other basic characteristics such as bitterness (IBU), original gravity, or color would also be useful.
Response: Thank you for your kind guidance. The added information was marked by red in part 2.1 of our revised version. The alcohol contents of sample A, B and C were 4%, 3%, and 4%, respectively.
Comment 5: Line 166: It might be helpful to include a brief explanation of how the OAVs of the flavor compounds were calculated, especially for clarity and completeness.
Response: Special thanks to you for your good comments and suggestions. The added information was marked by red in part 2.6 of our revised version. The OAVs of flavor compounds were calculated according to the ratio of the target odorant concentration to their threshold.
Comment 6: Line 179: Please include the standard or guideline that was followed for sensory analysis.
Response: Many thanks for your patient review and kind guidance. The revised part was marked by red in our revised version.
Comment 7: Lines 190-198: Could you please clarify how the concentrations of the specific compounds to be used in the validation experiments were determined?
Response: Special thanks for your patient review and kind guidance. We were willing to clarify it. We determined the concentrations of the specific compounds in the validation experiments according to the quantitative results of three samples. Specifically, the concentrations of the first group of compounds were less than twice their minimum concentrations among the three samples, and the concentrations of the second group of compounds were greater than twice their maximum concentrations among the three samples.
Comment 8: Line 199: Please provide a more detailed explanation of the INSENT taste analysis system used in the study.
Response: Thank you for your patient reading and kind advices. INSENT taste analysis system is a taste analysis system based on bionics principles, which belongs to the Electronic Tongue technology. It simulates the human taste perception mechanism through a multi-channel sensor array and can specifically respond to basic taste substances such as bitterness, sweetness, umami, saltiness and sourness. Compared with sensory evaluation, the electronic tongue has advantages such as no subjective deviation and strong repeatability. The added information was marked by red in our revised version.
Comment 9: Table 1: Since the aroma intensity is represented by the average result of panelists, the values in the table should be given as the average score and standard deviation. The table cannot contain empty cells.
Response: Thanks for your patient comments and suggestions. The added information was marked by red in our revised version. The values in the table were the average results after taking an integer and the standard deviations added were similar, because the three panelists recorded the scores in Osme as an integer with an interval of 0.5 between 0-5.
Comment 10: Table 2: Please clarify why the OAV values in Table 2 are presented as whole numbers, as this is not necessary. We encourage you to search for and include odor threshold values for all (or as many as possible) compounds. These values can also be found in studies related to wine, spirits, or other matrices (not necessarily related to beer). Including this information would enhance the discussion and improve the overall quality of the manuscript.
Response: Thank you very much for your patient review and kind suggestions. The revised part was marked by red in our revised manuscript. We have searched for and revised odor threshold values in different matrix solutions according to your suggestions.
Comment 11: Figure 3 should be improved in terms of clarity and presentation. Additionally, since you had 16 sensory panelists, it is possible to perform statistical comparisons for each evaluated parameter. Please consider including indicators of significant differences among samples in the figure. This would strengthen the interpretation of the sensory data.
Response: Thank you for your patient reading and kind advices. The three samples were lager beer, and most of the characteristics are similar among them. The figure 3 was improved with the mark of significant differences among samples according to your suggestions, and the revised text was marked by red in our revised manuscript.
Reviewer 3 Report
Comments and Suggestions for Authors
Manuscript foods-3598997
“ Characterization of key factors associated with flavor characteristics in lager based on flavor matrix”
The manuscript is clear and well structured. It mainly discusses the main results of the identification and validation of the main aroma compounds in lager beer and their sensory effects, with supporting tables and figures referenced throughout. It is an interesting work for the beer world.
However, there are some aspects that need to be corrected and improved, as follows:
Title:
“Characterization of key factors associated with flavor characteristics in lager beer based on flavor matrix”
- In the title and throughout the text when lager appears, lager beer should be included.
Abstract :
The abstract concisely summarizes the study’s aims, methods, key results, and implications.
Introduction:
The introduction provides background on the beer industry, especially in China, and the importance of flavor compounds in lager. It reviews previous research and sets out the study’s objectives.
- In line 41 update data according to Liu et al., (2017).
Liu, L., Wang, J., Rosenberg, D., Zhao, H., Lengyel, G., & Nadel, D. (2018). Fermented beverage and food storage in 13,000 y-old stone mortars at Raqefet Cave, Israel: Investigating Natufian ritual feasting. Journal of Archaeological Science: Reports, 21, 783-793.
- In lines 56-57 briefly describe what are the differences in terms of production and characteristics of lager and ale beers with references.
Materials and Methods:
This section describes the samples, reagents, analytical techniques (LLE, SAFE, HS-SPME, GC×GC-MS, GC-O-MS), sensory analysis, statistical methods, and validation experiments in detail.
- In line 124 check degree Celsius and in other places in the text substitute ºC.
Results
- The legends of the tables and figures should be more informative and specify what they consist of. For example, in Figure 4 it should be explained what each bubble is, although it is specified in the text.
Discussion?
This is not a discussion, this section summarises the main findings, methodological strengths and implications for improving lager beer flavour, so this is more like conclusions. I suggest a Results and Discussion section and a Conclusions section
Author Response
Comment 1: “Characterization of key factors associated with flavor characteristics in lager based on flavor matrix”
The manuscript is clear and well structured. It mainly discusses the main results of the identification and validation of the main aroma compounds in lager beer and their sensory effects, with supporting tables and figures referenced throughout. It is an interesting work for the beer world.
However, there are some aspects that need to be corrected and improved.
Response: Thank you for your patient review and kind guidance. Your suggestions are very valuable, which will help us to improve the quality of this text. We will revise it in red carefully according to your suggestions.
Comment 2: “Characterization of key factors associated with flavor characteristics in lager beer based on flavor matrix”
- In the title and throughout the text when lager appears, lager beer should be included.
Response: Many thanks for your patient review and kind suggestions. We have modified the expression in the title and full text and they have been marked in red.
Comment 3: - In line 41 update data according to Liu et al., (2017).Liu, L., Wang, J., Rosenberg, D., Zhao, H., Lengyel, G., & Nadel, D. (2018). Fermented beverage and food storage in 13,000 y-old stone mortars at Raqefet Cave, Israel: Investigating Natufian ritual feasting. Journal of Archaeological Science: Reports, 21, 783-793.
Response: Thank you for your kind suggestions. I agree with your points. The revised part and the added reference were marked in red in our revised manuscript.
Comment 4: In lines 56-57 briefly describe what are the differences in terms of production and characteristics of lager and ale beers with references.
Response: Special thanks to you for your good comments and suggestions. We have added the description about the characteristics of lager and ale beers with references. The revised part and the added reference were marked in red in our revised manuscript.
Comment 5: - In line 124 check degree Celsius and in other places in the text substitute ºC.
Response: Many thanks for your patient reading and advice. We have revised the degree Celsius and they have been marked in red.
Comment 6: -The legends of the tables and figures should be more informative and specify what they consist of. For example, in Figure 4 it should be explained what each bubble is, although it is specified in the text.
Response: Thanks for your patient comments and suggestions. We have revised the legends of the tables and figures. The revised parts were marked in red.
Comment 7: This is not a discussion, this section summarises the main findings, methodological strengths and implications for improving lager beer flavour, so this is more like conclusions. I suggest a Results and Discussion section and a Conclusions section.
Response: Thank you for your patient review and kind guidance. The changed parts were marked by red in our revised manuscript. We have revised the part 4 according to your advice.